# Uncovering Pathways Highly Correlated to NUE through a Combined Metabolomics and Transcriptomics Approach in Eggplant

**DOI:** 10.3390/plants11050700

**Published:** 2022-03-04

**Authors:** Antonio Mauceri, Meriem Miyassa Aci, Laura Toppino, Sayantan Panda, Sagit Meir, Francesco Mercati, Fabrizio Araniti, Antonio Lupini, Maria Rosaria Panuccio, Giuseppe Leonardo Rotino, Asaph Aharoni, Maria Rosa Abenavoli, Francesco Sunseri

**Affiliations:** 1Department Agraria, University Mediterranea of Reggio Calabria, 89122 Reggio Calabria, Italy; miyassa.aci@unirc.it (M.M.A.); antonio.lupini@unirc.it (A.L.); mpanuccio@unirc.it (M.R.P.); francesco.sunseri@unirc.it (F.S.); 2CREA—Research Centre for Genomics and Bioinformatics, 26836 Montanaso Lombardo, Italy; laura.toppino@crea.gov.it (L.T.); giuseppeleonardo.rotino@crea.gov.it (G.L.R.); 3Department of Plant and Environmental Sciences, Weizmann Institute of Science, Rehovot 7610001, Israel; sayantan.panda@weizmann.ac.il (S.P.); sagit.meir@weizmann.ac.il (S.M.); asaph.aharoni@weizmann.ac.il (A.A.); 4Institute Bioscience and Bioresources—National Research Council CNR, 90129 Palermo, Italy; francesco.mercati@ibbr.cnr.it; 5Department of Agricultural and Environmental Sciences—Production, Territory, Agroenergy, University of Milano, 20133 Milan, Italy; fabrizio.araniti@unimi.it

**Keywords:** *Solanum melongena* L., primary metabolites, GC-MS, glycoalkaloids, UPLC-qTOF-MS, RNA-seq, nitrogen-use efficiency

## Abstract

Nitrogen (N) fertilization is one of the main inputs to increase crop yield and food production. However, crops utilize only 30–40% of N applied; the remainder is leached into the soil, causing environmental and health damage. In this scenario, the improvement of nitrogen-use efficiency (NUE) will be an essential strategy for sustainable agriculture. Here, we compared two pairs of NUE-contrasting eggplant (*Solanum melongena* L.) genotypes, employing GC-MS and UPLC-qTOF-MS-based technologies to determine the differential profiles of primary and secondary metabolites in root and shoot tissues, under N starvation as well as at short- and long-term N-limiting resupply. Firstly, differences in the primary metabolism pathways of shoots related to alanine, aspartate and glutamate; starch, sucrose and glycine; serine and threonine; and in secondary metabolites biosynthesis were detected. An integrated analysis between differentially accumulated metabolites and expressed transcripts highlighted a key role of glycine accumulation and the related *glyA* transcript in the N-use-efficient genotypes to cope with N-limiting stress. Interestingly, a correlation between both sucrose synthase (*SUS*)- and fructokinase (*scrK*)-transcript abundances, as well as D-glucose and D-fructose accumulation, appeared useful to distinguish the N-use-efficient genotypes. Furthermore, increased levels of L-aspartate and L-asparagine in the N-use-efficient genotypes at short-term low-N exposure were detected. Granule-bound starch synthase (*WAXY*) and endoglucanase (*E3.2.1.4*) downregulation at long-term N stress was observed. Therefore, genes and metabolites related to these pathways could be exploited to improve NUE in eggplant.

## 1. Introduction

Soil-N availability is one of the most important factors limiting worldwide plant growth and productivity. Nitrogen limitation leads to many functional damages, inducing alterations in physiological, biochemical, and molecular processes such as photosynthesis, respiration, ion uptake and translocation, carbon metabolism, and senescence [1,2]. Over the last four decades, N employed in fertilization has dramatically increased in order to maximize crop yield and consequently meet food demands [3]. However, excessive use of N determines negative effects on the environment, economy, and human health [4]. Thus, nitrogen-use-efficiency (NUE) improvement in crop plants, together with low-N-fertilizer input and best-management practices, could represent strategies to limit the negative impact of agriculture on the environment and human health [5].

NUE, defined as “the grain yield per unit of N available in the soil”, is a complex trait under physiological, biochemical, and genetic control [6,7]. Efforts have been made to identify molecular mechanisms underlying NUE to improve this complex trait by conventional breeding programs in several crops [8]. Recently, metabolomics, transcriptomics, and proteomics are becoming valuable tools in model and crop species to understand changes in biological processes, genes, and chemical composition of primary and secondary metabolites involved in stress responses, including N deficiency [9,10,11].

NUE has been mainly explored in several crop plants, such as rice, wheat, and maize [12], but should be relevant also for *Solanaceae* crops due to their high dependence in N fertilizers [13]. Limited information on NUE is available on eggplant (*Solanum melongena* L.), the third most important vegetable crop (following tomato and potato) that is cultivated worldwide, mainly in Asian territories [13]. Recently, two NUE-contrasting genotypes have been identified in hydroponic and greenhouse experiments, under low-N conditions [14]. Transcriptome analysis, in both shoot and root tissues, highlighted differentially expressed genes (DEGs) related to NUE traits after short- and long-term N-stress exposure. In detail, DEGs involved in the light-reaction pathway, the response to inorganic substances, abiotic stimulus, and cellular response to N starvation, together with several putative transcription factors (TFs) were upregulated in the N-use-efficient genotypes [15].

Here, we use a metabolomics approach to examine the metabolic profiles of NUE-contrasting genotypes [14], with particular emphasis to organic acids, amino acids, sugars, and secondary metabolites, mainly on nitrogen-containing ones such as glycoalkaloids that previously have been considered. In particular, glycoalkaloids (GAs), a class of nitrogen-containing secondary metabolites, commonly occur in the *Solanaceae* family (tomato, potato, and eggplant). Although considered toxic for human health, they exhibit a wide range of pharmacological properties, including anticancer activity; whereas in plants, they play an important defense role against pests [16,17]. Interestingly, plants change their metabolic profiles under N stress, according to the starvation period, genotypes, and tissues [18,19]. In particular, N- and C-containing metabolites (including organic acids) are strictly correlated with plant biomass and growth [20].

To identify key pathways involved in N metabolism in eggplant, an integration between metabolomics and transcriptomics data was carried out. Our findings shed light on genes and metabolites reprogrammed in N-use-efficient genotypes compared to inefficient genotypes under N-limiting resupply, providing valuable knowledge to develop strategies for improving NUE in eggplant.

## 2. Results

### 2.1. Metabolite Detection in Contrasting NUE Genotypes at Different Resupply Time Intervals

Root and shoot extracts derived from high (AM222 and 67-3)- and low (305E40 and AM22)-NUE eggplant genotypes were profiled using GC-MS (mainly primary, polar metabolites) and high-resolution MS (HRMS; UPLC-qTOF-MS; mainly secondary, semi-polar metabolites) to examine the plant response to N starvation as well as short- and long-term N-limited resupply. In both tissues, collected at T_0_ (after 2 days N-free solution), T_1_ and T_2_ (1 and 16 days after NO_3_^−^ resupply), 33 (through GC-MS) and 29 (through UPLC-qTOF-MS) metabolites were annotated in high confidence. GC-MS analysis results in the identification of three main primary metabolite classes including amino acids (17), organic acids (11), and sugars (5) (Appendix A). In HRMS analysis, we focused on nitrogen-containing metabolites, mainly glycoalkaloids (Appendix A).

### 2.2. Multivariate Statistical Analysis of Eggplant Metabolite

The PCA model performed on root data took into account the two main components, PC1 vs. PC2, which explained 31.6% and 9.4% of the total variance (41%), respectively. The analysis did not clearly separate the samples (Figure 1A). Loading plots of metabolites according to different times and genotypes in root are shown in Appendix A. Conversely, in shoot, PC1 and PC2 explained 20.4% and 15% of the total variance (35.4%), respectively, pointing out proximity between time sampling T_0_ and T_1_, while T_2_ appeared more distinguished (Figure 1B). The more N-use-efficient genotypes, AM222, 305E40, and 67-3, showed distinct metabolic profiles (in T_2_); by contrast, the AM22 did not show any significant difference across the time of sampling (Figure 1B, Appendix A).

To maximize the variance, a partial least-squares discriminant analysis (PLS-DA) was performed on the same datasets. In root, the PLS-DA permutation test showed an empirical *p*-value ≥ 0.05; therefore it was not considered. In shoot, the PLS-DA model defined a clearer separation, on which components 1 and 2 explained 13.9% and 12.1% of the total variance, respectively. At all sampling times, the replicates of each genotype were closely grouped (apart from the AM222 samples at T_1_; Figure 2). AM22 displayed reduced metabolic changes across sampling times (T_0_, T_1_, and T_2_) compared to the other genotypes. Notably, at T_2_, the high-NUE genotypes (i.e., AM222 and 67-3), appeared well-separated from 305E40 and AM22 (the lower-NUE genotypes; Figure 2A). The variable importance in projection (VIP) indicated L-glutamine, L-isoleucine, L-cysteine, myo-inositol, L-phenylalanine, solamargine isomer (M869T830), shikimic, L-glutamic, oxalic, and succinic acids as the most discriminant metabolites among the genotype-profile scores (Figure 2B, Appendix A). More specifically, at T_0_, the low-NUE genotype AM22 showed high levels of L-glutamic acid, L-cysteine, and L-isoleucine, while 305E40 showed high levels of L-glutamine. Conversely, in the high-NUE genotypes, 67-3 exhibited high levels of the unknown glycoalkaloid M1003T1070, while AM222 the solamargine isomer (M869T830). At T_1_, AM22, and 305E40 accumulated shikimic acid and L-phenylalanine, respectively. Furthermore, 67-3 and AM222 showed high levels of oxalic acid and the solamargine isomer (M869T830), as well as L-cysteine. Finally, at T_2_, AM22 accumulated succinic acid, L-glutamic acid, and myo-inositol, while 305E40 exhibited high levels of solamargine isomer (M869T830), L-isoleucine, and L-leucine (Figure 2B, Appendix A).

### 2.3. Root and Shoot Metabolites in Eggplant Genotypes under Low N Supply

The analysis of variance (ANOVA), comparing the overall variation among the identified metabolites (*p* < 0.05), is provided in Appendix A. In the root tissue, differences were mainly observed in secondary metabolites while only the myo-inositol was changed among the primary ones. In particular, at T_0_, the N-use-inefficient genotype AM22 showed significant accumulation of secondary metabolites, mainly furostanol and the soladulcine A, together with myo-inositol (Figure 3A), while at T_1_ and T_2_, the secondary metabolites were gradually reduced. Interestingly, at T_1_, the efficient genotype AM222 displayed an increase in solasonine (S) (M885T1016), solamargine isomer (M869T830), solasonine isomer (M885T1031), and hydroxy dihydro solasonine (M903T687) (Figure 3B). In contrast to root, the shoot tissue showed differences in both primary and secondary metabolites (Figure 4). D-glucose and D-fructose showed the highest levels among sugars, while L-aspartic and L-glutamic acids, L-isoleucine, L-proline, L-glutamine, L-phenylalanine, L-asparagine, L-threonine, L-cysteine, and glycine showed highest levels among the amino acids. Finally, L-malic, oxalic, succinic, shikimic, glyceric, and fumaric acids appeared significantly variable among the organic acids comparing sampling times and genotypes (Figure 4). The glycoalkaloids solamargine isomer (M869T830), hydroxy dihydro solamargine (M887T705), hydroxy-solamargine isomer 1 (M885T703), unsaturated unidentified glycoalkaloid (UGA; M862T924), and unsaturated malonyl furostanol type saponin (M1118T1173) were differentially accumulated (Figure 4).

More specifically, the N-use-efficient genotype AM222 showed the highest levels of L-malic and shikimic acids as well as D-glucose at N starvation (T_0_); conversely the N-use -inefficient AM22 exhibited the highest levels of fumaric and glyceric acids, several amino acids, and secondary metabolites (Figure 4A). At T_1_, AM222 showed the highest level of L-serine, L-valine, and L-leucine, while AM22 showed the highest levels of D-glucose and D-fructose as well as fumaric and shikimic acids (Figure 4B). Finally, at T_2_, AM222 exhibited the highest level of shikimic acid and D-glucose as well as secondary metabolites, while oxalic, succinic, L-aspartic, and L-glutamic acids accumulated more in AM22 than AM222 (Figure 4C).

### 2.4. Comparative Changes in the Primary Metabolite Pathways in Shoot

Significant different metabolic pathways (*p* < 0.05, FDR < 0.05, and impact >0.2) were identified by pairwise comparisons among genotypes at the same sampling time, among which are alanine, aspartate and glutamate (impact 0.65); glycine, serine and threonine (impact 0.54); phenylalanine (impact 0.47); starch and sucrose (impact 0.39); and glyoxylate and dicarboxylate (impact 0.28) (Table 1).

To identify differently accumulated metabolites in each pathway, a comparison between AM22, the most N-use-inefficient, and all other genotypes was performed (Appendix A; all the other comparisons are given in Appendix A). At T_0_, 67-3 showed a significant reduction in L-aspartate, while 305E40 exhibited a significant lower L-glutamate and succinate content compared to AM22 (Appendix A). At T_1_, AM222 and 67-3 showed a higher content in L-aspartate and a lower level in L-glutamate; by contrast, 305E40 evidenced a decrease in L-glutamine, L-glutamate, and succinate content compared to AM22 (Appendix A). At T_2_, AM222 significantly differed in starch and sucrose metabolism compared to AM22, with a higher content in sucrose and D-glucose (Appendix A). By contrast, 67-3 showed a reduction in L-asparagine, L-alanine, L-glutamate, and succinate content (Appendix A). Finally, 305E40 displayed an increase in D-glycerate and L-threonine content, as well as a decrease in glycine as compared to AM22 (Appendix A).

### 2.5. Metabolite- and Transcript-Correlation Analysis

To investigate the pathways most significantly affected by N-limiting condition in the NUE-contrasting eggplants, we performed an integrated analysis of metabolomics and transcriptomics datasets [15] obtained by the same experimental setup. KEGG database was used to annotate the eggplant genes in the same pathway. Thus, 121 genes including 17, 13, 33, 21, 7, 26, and 4 genes, respectively, from the aminoacyl-tRNA biosynthesis (ko00970); the alanine, aspartate, and glutamate metabolism (ko00250); the starch and sucrose metabolism (ko00500); the glycine, serine, and threonine metabolism (ko00260); the phenylalanine metabolism (ko00360); the glyoxylate and dicarboxylate metabolism (ko00630); and the isoquinoline alkaloid biosynthesis (ko00950, related to the secondary metabolism), were identified (Appendix A).

To build a correlation network between metabolites and gene transcripts, a Pearson’s correlation on a 121-genes and 39-metabolite dataset from both tissues was performed. Seventy (70) correlated variables (Appendix A) were highlighted with a correlation coefficient ≥0.70 and ≤−0.70 and a *p*-value < 0.05 (Appendix A). These 70 correlated variables were then subjected to pathway-enrichment analysis, which identified the most significant metabolic pathway (impact ≥0.6) affected by treatments in the glycine, serine, and threonine; starch and sucrose; and glyoxylate and dicarboxylate metabolism pathways (Appendix A). Metabolite and gene differences and their correlations were visualized through a heatmap for each genotype and time sampling (T_0_, T_1_ and T_2_). The N-use-efficient (AM222 and 67-3) and inefficient (305E40 and AM22) genotypes clustered separately in shoot at T_0_ and T_1_, while a clustering between NUE-contrasting genotypes was also observed in root at T_1_. At T_2_, in both tissues, AM222 and AM22, the more contrasting genotypes for NUE clustered distinctly (Appendix A).

### 2.6. Genotype Clustering and Responses to Nitrogen Starvation

At T_0_, in shoot the N-use-efficient genotypes, AM222 and 67-3, clustered due to differences in transcript abundances of sucrose synthase (*SUS*; SMEL_007g277290.1), a component of starch and sucrose metabolism; glycine hydroxy methyltransferase (*glyA*; SMEL_005g228290.1; synonym of *shmt1*, serine hydroxymethyltransferase 1) and dihydrolipoamide dehydrogenase (*DLD*; SMEL_005g227630.1) components of glycine, serine, and threonine metabolism; and malate dehydrogenase (*MDH1*; SMEL_009g332450.1) and isocitrate lyase (*aceA*; SMEL_007g288630.1) which are involved in glyoxylate and dicarboxylate metabolism. A higher accumulation in D-fructose in AM222 was also observed (Appendix A). By contrast, AM22 and 305E40, the N-use-inefficient genotypes, accumulated more transcripts of the polyphenol oxidase (*PPO*; SMEL_000g064350.1, isoquinoline alkaloid biosynthesis), two different sucrose synthase isoforms (*SUS*; SMEL_007g277310.1 and SMEL_012g382160.1), and 4-alpha-glucanotransferase (*malQ*; SMEL_007g279110.1). They further showed an upregulation in the phenylalanine decarboxylase (*AADC*; SMEL_008g301360.1) and phenylalanine ammonia-lyase (*PAL*; SMEL_005g230690.1) involved into phenylalanine metabolism. AM22 and 305E40 showed also a higher accumulation of fumaric acid as well as all the glycoalkaloids (Appendix A).

At T_0_, in root, the glyoxylate/succinic semialdehyde reductase (*GLYR*; SMEL_000g004100.1), granule-bound starch synthase (*WAXY*; SMEL_000g036450.1) and glutamate decarboxylase (*GAD*; SMEL_001g150880.1) appeared upregulated in AM22 together with a higher accumulation in myo-inositol compared to the other genotypes; by contrast in 305E40, the amidophosphoribosyltransferase (*purF*; SMEL_001g116660.1), threonine synthase (*thrC*; SMEL_003g 197730.1), glucan endo-1,3-beta-glucosidase 4 (*GN4*; SMEL_001g152910.1) and isocitrate lyase (*aceA*; SMEL_007g288630.1) were upregulated compared to the others, accompanied by a higher accumulation of secondary metabolites. Furthermore, a strong downregulation of starch synthase (*glgA*; SMEL_000g085440.1) and a lower accumulation of L-alanine was exhibited (Appendix A).

### 2.7. Genotype Clustering and Short-Term Responses to Low Nitrogen Supply

At T_1_, in shoot, two distinguishable clusters between the N-use-efficient genotypes and the inefficient ones were generated (Appendix A). The AM222 and 67-3 showed an upregulation of the granule-bound starch synthase (*WAXY*; SMEL_000g036450.1) and endoglucanase (*E3.2.1.4*; SMEL_003g177010.1) involved in the starch and sucrose metabolism, as well as phenylalanine decarboxylase (*AADC*; SMEL_005g228290.1) and glycine hydroxy methyltransferase (*glyA*; SMEL_000g004730.1) comprised in the glyoxylate and dicarboxylate metabolism (Appendix A).

Otherwise, the AM22 and 305E40 evidenced an upregulation of many genes involved in different pathways (Appendix A). In particular, the glycine dehydrogenase (*GLDC*; SMEL_008g307820.1; glyoxylate and dicarboxylate metabolism) transcripts appeared more expressed together with a higher accumulation in D-glucose and sucrose; the isocitrate lyase (*aceA*; SMEL_007g288630.1; glyoxylate and dicarboxylate metabolism) was upregulated together with myo-inositol; the alanyl-tRNA and aspartyl-tRNA synthetases (*AARS*; SMEL_001g115360.1 and *DARS2*; SMEL_000g033540.1 in the aminoacyl-tRNA biosynthesis) were upregulated, accompanied by a higher accumulation in fumaric acid and L-alanine. Interestingly, these genotypes confirmed a higher accumulation of all the glycoalkaloids compared to the N-use-efficient genotypes as already observed at T_0_ (Appendix A).

At T_1_, two distinct clusters between the NUE-contrasting genotypes were confirmed in root. The efficient genotypes showed a significant upregulation of the endoglucanase (*E3.2.1.4*; SMEL_003g177010.1) and sucrose synthase isoform (*SUS*; SMEL_003g277290.1) involved in the starch and sucrose metabolism (Appendix A). By contrast, the (S)-2-hydroxy-acid oxidase (*HAO*; SMEL_004g202790.1; glyoxylate and dicarboxylate metabolism) resulted down-regulated. In the N-use-inefficient genotype 305E40, another sucrose synthase isoform (*SUS*; SMEL_012g382160.1), and trehalose 6-phosphate synthase/phosphatase (*TPS*; SMEL_001g151160.1), involved into the starch and sucrose metabolism, were upregulated concurrently with a higher accumulation of several secondary metabolites (Appendix A). It also exhibited a higher glucan endo-1,3-beta-glucosidase 4 (*GN4*; SMEL_001g152910.1) and 4-alpha-glucano transferase (*malQ*; SMEL_007g279110.1) transcripts abundances in the starch and sucrose metabolism as well as the aspartate-semialdehyde dehydrogenase (*asd*; SMEL_001g151200.1; glycine, serine, and threonine metabolism) and amidophosphoribosyl-transferase (*purF*; SMEL_001g116660.1; alanine, aspartate, and glutamate metabolism) (Appendix A).

### 2.8. Genotype Clustering and Long-Term Responses to Low Nitrogen Supply

At T_2_, in shoot, AM222 distinctly clustered from the other genotypes. It included higher granule-bound starch synthase (*WAXY*; SMEL_000g036450.1) transcript levels together with a significant higher D-glucose accumulation; by contrast, a downregulation of (S)-2-hydroxy-acid oxidase (*HAO*; SMEL_007g292380.1) histidyl-tRNA synthetase (*HARS*; SMEL_003g195740.1), which correlated with a high accumulation in the primary metabolites D-fructose, shikimic, and quinic acids was observed. In addition, AM222 showed a high polyphenol oxidase (*PPO*; SMEL_008g312510.1) and phenyl alanine ammonia-lyase (*PAL*; SMEL_005g230690.1) expression in the phenylalanine metabolism as well as two and three genes in the glyoxylate and dicarboxylate and starch and sucrose metabolisms, respectively. In this last pathway, two sucrose synthase (*SUS*; SMEL_007g277310.1 and SMEL_007g277290.1) isoforms and the fructokinase (*scrK*; SMEL_006g265210.1) appeared upregulated in AM222 and 67-3 (Appendix A).

AM22 and 305E40 were mainly distinguishable from the N-use-efficient genotypes for a large cluster of correlation, in which a higher catalase (*CAT*; SMEL_000g061370.1), (S)-2-hydroxy-acid oxidase (*HAO*; SMEL_007g292380.1), glycine dehydrogenase (*GLDC*; SMEL_008g307820.1), and isocitrate lyase (*aceA*; SMEL_007g288630.1) transcript abundances belonging to the glyoxylate and dicarboxylate metabolism were observed, together with a higher accumulation in the fumaric and dehydroascorbic acids as well as in L-serine, L-alanine, and L-asparagine (Appendix A).

At T_2_, in root, AM222 showed an upregulation of the aspartyl-tRNA synthetase (*DARS2*; SMEL_000g033540.1), glucose-1-phosphate adenylyl transferase (*glgC*; SMEL_007g292070.1) and glyoxylate/succinic semialdehyde reductase (*GLYR*; SMEL_000g004100.1), accompanied by a higher accumulation in sucrose and amino acids L-alanine and glycine. It also showed a higher expression in the phenylalanine ammonia-lyase (*PAL*; SMEL_ 005g230690.1) sucrose synthase (*SUS*; SMEL_007g277290.1), two glycine hydroxy methyltransferase isoforms (*glyA*; SMEL_005g228290.1 and SMEL_005g241460.1), and a glycine cleavage system H protein (*gcvH*; SMEL_000g091530.1) belonging to glycine, serine, and threonine metabolism, accompanied by a higher accumulation in the amino acid L-asparagine compared to the other genotypes (Appendix A).

Otherwise, AM22 showed a distinguishable cluster of correlation including a strong downregulation of the starch synthase (*glgA*; SMEL_000g085440.1) and aspartyl-tRNA synthetase (*DARS2*; SMEL_ 000g033540.1) together with a very low sucrose accumulation. It also displayed a higher threonine synthase (*thrC*; SMEL_003g197730.1) and amidophosphoribosyltransferase (*purF*; SMEL_001g116660.1) transcript abundance compared to the other genotypes (Appendix A).

### 2.9. Implementing a Simplified Modeling Scheme

Overall, genes and metabolites that allowed for discrimination of the N-use-efficient vs. inefficient genotypes mainly belonged to the glycine, serine, and threonine (Appendix A); glyoxylate and dicarboxylate (Appendix A); and the starch and sucrose metabolism pathways (Appendix A). In particular, at T_0_, N-use-efficient genotypes showed a higher *asd* (i.e., aspartate-semialdehyde dehydrogenase) transcript abundance in the glycine, serine, and threonine metabolism. This gene forms a branch with the metabolic pathway making lysine, methionine, leucine, and isoleucine from aspartate. The same genotypes also showed a glycine hydroxymethyltransferase (*glyA*; SMEL_005g228290.1) downregulation, useful for the concurrent conversions of L-serine to glycine and tetrahydrofolate (THF) to 5,10-methylenetetrahydrofolate (Appendix A). Conversely, at T_1_ and T_2_, *asd* and *glyA* showed an inverted expression trend in the efficient genotypes, resulting in a higher *glyA* and lower *asd* expression compared to the inefficient ones (Appendix A). As a consequence, a higher level of glycine as well as of the glycine-cleavage-system H protein (*gcvH*) and dihydrolipoamide dehydrogenase (*DLD*) expression that regulates the glycine concentration and cell energy metabolism, respectively, was observed. By contrast, a higher accumulation of serine and glyceric acid, together with the upregulation of *AGXT* (alanine-glyoxylate transaminase) and *GLCD* (glycine dehydrogenase) was detected in the N-use-inefficient genotypes (Appendix A).

In the glyoxylate and dicarboxylate metabolism, the N-use-efficient genotypes showed *aceA* (isocitrate lyase) upregulation in the glyoxylate cycle, as well as a higher accumulation in *HAO* [(S)-2-hydroxy-acid oxidase] and *CAT* (catalase) transcripts involved in the photorespiration cycle, and a low accumulation of *glyA* compared to the inefficient ones at N starvation (T_0_) (Appendix A). At T_1_, the N-use-efficient genotypes showed a significant downregulation of some transcripts involved in the glyoxylate cycle, such as *CS* (citrate synthase), *ACO* (aconitate hydratase), and *aceA* (isocitrate lyase), as well as *AGXT* (alanine-glyoxylate transaminase) and *HPR2_3* (glyoxylate/hydroxypyruvate reductase) involved in the photorespiration cycle; by contrast, *CAT* (catalase) and *glyA* (glycine hydroxymethyltransferase) resulted in upregulation (Appendix A).

Otherwise, at T_2_, the N-use-efficient genotypes exhibited a significant upregulation of *ACO* and *glyA* transcripts and a higher accumulation in glycine; by contrast, a downregulation of *MDH* (malate dehydrogenase) and *aceA* involved in the glyoxylate cycle, as well as *HAO*, *CAT*, *AGXT*, *rbcL* (ribulose-bisphosphate carboxylase large chain), and *GLDC* (glycine dehydrogenase) in the photorespiration cycle, together with a consistent reduction of glyceric acid, were observed (Appendix A). In the starch and sucrose metabolism, at T_0_, the N-use-efficient compared to the inefficient genotypes showed a significantly higher expression of *SUS* (sucrose synthase) and *E3.2.1.4* (endoglucanase), while the *INV* (invertase), *GN4* (glucan endo-1,3-beta-glucosidase 4) and *TPS* (trehalose 6-phosphate synthase/phosphatase) resulted in downregulation (Appendix A). At T_1_, the N-use-efficient genotypes exhibited a higher transcript accumulation in the *WAXY* and *endoglucanase*, which are upstream amylose and cellobiose biosynthesis, together with the downregulation of *SUS*, *scrK* (fructokinase), and *TPS* genes, which resulted in a lower accumulation of sucrose (Appendix A). Conversely, at T_2_, in the N-use-efficient genotypes we observed an upregulation of *SUS* and *scrK*, as well as a higher accumulation of D-glucose and D-fructose; meanwhile, *WAXY*, *endoglucanase*, and *glgC* (glucose-1-phosphate adenyltransferase) appeared strongly downregulated (Appendix A).

## 3. Discussion

In this study, combined metabolomics and transcriptomics analysis in two pairs of NUE-contrasting eggplant genotypes, AM222, 67-3 (high NUE); and 305E40, AM22 (low NUE) was performed, in root and shoot, under short- and long-term N-limiting conditions. Plant responses to low N resupply in NUE-contrasting genotypes are of particular interest to dissect the key molecular mechanisms underlying this complex trait, to identify the critical steps controlling NUE, and to provide new insights for breeding programs to improve NUE. Here, the base for deciphering metabolites and gene-correlation networks to facilitate NUE improvement in eggplant was provided, as previously reported in other crops [21,22].

### 3.1. Variance and Pathway Analysis

At all the sampling times, root showed similar primary metabolite profiles regardless of genotype, while variations were mostly detected in the secondary metabolites. By contrast, in shoot, variations in primary metabolites were revealed across sampling times (except for the genotype AM22) and between genotypes when compared at the same sampling time. To explore these differences, all the possible pairwise comparisons were performed between the most N-use-inefficient genotype, AM22, and the other genotypes, allowing us to highlight the affected pathways in this tissue.

Under N starvation (T_0_), the content of most amino acids was reduced in the N-use-efficient genotypes, mainly L-glutamine. Low levels of N-containing metabolites, such as glutamate and glutamine, and C-containing compounds were already observed under N starvation together with an elevated level of organic acids, suggesting their utilization to build macromolecules [20]. At T_1_, L-aspartate and L-asparagine, belonging to the alanine, aspartate, and glutamate pathway, resulted in an increase in N-use-efficient genotypes. In detail, L-asparagine, the amino acid with the highest N:C ratio, plays an important role in N transport and storage through the vascular system, or alternatively accumulates in response to stress, contributing to osmotic-pressure maintenance [23]. Finally, after a long-term N-limiting condition (T_2_), AM222 showed a higher accumulation in D-glucose and sucrose (starch and sucrose metabolism), while 67-3 evidenced a reduction in all the metabolites in the alanine, aspartate, and glutamate metabolism, but a slight L-glutamine accumulation. Sugars frequently play an important role as osmoprotectors, as membrane stabilizers, in stress buffering [24], as well as in the regulation of growth, photosynthesis, carbon partitioning, carbohydrate and lipid metabolism, protein synthesis, and gene expression [25]. Interestingly, the most N-use-efficient genotype AM222 compared to the other genotypes showed higher accumulation of phenylalanine and citric acid that were reported to confer tolerance to abiotic stress, improving growth and yield in many crop species [26]. By contrast, an increase in proline and alanine in the N-use-inefficient genotype AM22 could indicate prolonged stress due to the unbalanced N supply [27]. Indeed, it is well-known that proline accumulates in plants subjected to environmental stress [28], maintaining osmotic balance and protecting cells against ROS under salt stress [29]. The biological significance of the alanine accumulation in plants is still controversial; indeed, its accumulation may be a N-storage mechanism, before restoring to an N-normal condition [30]. Thus, it is more difficult to explain the alanine increase in the N-use-inefficient genotypes. In the last decade, many efforts have been carried out to improve NUE through genetic engineering by overexpressing N-assimilation genes, among which includes the *alanine aminotransferase* (*AlaAT*) [31]. However, inconsistent results were obtained from transgenic plants evaluated under field conditions, suggesting that the overexpression of N-assimilation genes may cause metabolic imbalances [32,33].

### 3.2. Metabolite and Transcript Correlation Analysis

Metabolomics and transcriptomic data were analyzed by Pearson correlation analysis to identify genes and metabolites that concurrently distinguished N-use-efficient and inefficient eggplant genotypes. In accordance to Cavill et al. [34], our analyses integrated a subset of 121 genes and 39 metabolites (primary and secondary), resulting into 70 variables (47 transcripts and 23 metabolites). The correlation heatmaps between genes and metabolites showed different clusters in which AM222 and AM22 (the extreme NUE-contrasting genotypes) were always well-distinguished in both shoot and root.

Our findings suggest that differences between the two pairs of NUE-contrasting genotypes in secondary-metabolite biosynthesis; glyoxylate and dicarboxylate; glycine, serine, and threonine; and starch and sucrose metabolism pathways could be crucial for N-use efficiency in eggplant, in both the short and long term (T_1_ and T_2_). In agreement, DEGs and metabolic changes in amino acid, carbon, and nitrogen metabolism pathways were observed between two NUE-contrasting cotton genotypes in response to N starvation and resupply treatments [35]. These results showed an enrichment in the starch and sucrose metabolism, glycolysis/gluconeogenesis, and pentose phosphate pathways in N-use-efficient cotton genotypes, underlying that plant-energy budget as well as carbon and nitrogen metabolism and their balance are involved in the different NUE performances [35,36]. By contrast, in our experiments the N-use-inefficient genotypes showed a significant higher fumaric acid accumulation, previously observed in a starchless *pgm1* mutant [37]. This implied that fumaric acid and starch should serve as alternative carbon sinks for photosynthate, resulting in an effective higher N assimilation and *Arabidopsis* growth only when high N is available [38].

Interestingly, a significant PPOs upregulation at long-term low-N stress in AM222, the high-NUE genotype, was observed. Phenol-oxidizing enzymes are responsible of browning enzymatic reaction in post-harvest fruits and vegetables; although the PPOs’ native physiological functions in intact and undamaged plant cells are not still understood to date [39]. Recently, PPOs were reported to play different roles in response to plant stress in several species and may have an indirect role in photosynthesis [40]. By contrast, two different polyphenol oxidases (PPOs), an unsaturated malonyl solamargine and an unknown steroidal saponin, resulted in upregulation/accumulation in the N-use-inefficient eggplant genotypes, at both N-starvation (T_0_) and short-term low-N stress (T_1_).

### 3.3. Glycine, Serine, and Threonine Metabolism

Glycine is the main player of this pathway, which also includes the serine and betaine biosynthesis, which have been reported to be involved in plant responses to abiotic stress [41,42,43]. At T_0_, we observed in the N-use-efficient genotypes a very low level of *glyA* transcript encoding the enzyme, converting L-serine to glycine and tetrahydrofolate (THF). Furthermore, a higher aspartate-semialdehyde dehydrogenase (ASD) gene expression in trehalose biosynthesis sustains the formation of lysine and other amino acids from aspartate. Interestingly, under short- and long-term low-N stress (at T_1_ and T_2_), *glyA* (also named *shmt1*) inverted the expression trend, resulting in an increase in both AM222 and 67-3 compared to the N-use-inefficient genotypes. A high *shmt1* transcript level was reported, correlating with a reduced sensitivity to abiotic stress in *Arabidopsis* [42]. The higher *glyA* expression in the N-use-efficient genotypes resulted in very high glycine accumulation, mainly at T_2_, which could sustain the betaine formation, two primary metabolites that were reported to accumulate under abiotic stress in rice and several other plants [41,43]. More interestingly, the regulation of the glycine-concentration and the cell-energy metabolism appeared to be further guaranteed by a higher glycine-cleavage-system H protein (*gcvH*) and dihydrolipoamide dehydrogenase (*DLD*) expression in these genotypes.

By contrast, a higher accumulation of serine and glyceric acid (instead of glycine) in the N-use-inefficient genotypes appeared sustained by the *glyA* (=*shmt1*) downregulation together with a significant upregulation of *AGXT* (alanine-glyoxylate transaminase).

### 3.4. Glyoxylate and Dicarboxylate Metabolism

In plants, glyoxysomes frequently store lipids, and through the glyoxylate cycle, they are involved in the conversion of acetyl-CoA to succinate for the synthesis of carbohydrates. Under N starvation (T_0_), the N-use-efficient compared to inefficient genotypes exhibited a higher *aceA* transcript abundance. *AceA* encodes for isocitrate lyase, a key enzyme in the glyoxylate cycle that could play a pivotal role in energy metabolism for facing up stress as described by Yuenyong et al. [44]. The *HAO* (S)-2-hydroxy-acid oxidase) and *CAT* (catalase) upregulation in the same genotypes could also act on the glycolate–glyoxylate conversion for preventing the accumulation of glycolate and hydrogen peroxide at toxic levels as reported in maize [45].

Afterwards, at short low-N stress (T_1_), in the N-use-efficient genotypes an upregulation of *glyA* (*shmt1*), involved in the photorespiration cycle, was observed. The encoded enzyme catalyzes the interconversion of glycine (glyoxylate-derived) to serine and tetrahydrofolate (THF) acting as a carbon carrier. These compounds, accompanied to a higher *CAT* expression, could mitigate oxidative stress, driving an abiotic stress tolerance as already observed in *Arabidopsis* [46]. By contrast, *CS*, *ACO*, and *aceA* were downregulated in the efficient genotypes, and in particular, the inhibition of ACO activity was useful for plants to cope with oxidative stress, also correlating with cell death [47].

At the long-term N-stress (T_2_), *glyA* resulted in upregulation, together with a higher glycine accumulation in the N-use-efficient genotypes. Interestingly, this amino acid and its derivative, glycinbetaine, were more accumulated under abiotic stress in rice and other plant species [41,43]. By contrast, *aceA* and the malate dehydrogenase (MDH) genes downregulation resulted in an efficient redox activity of the mitochondrial matrix as reported in *Arabidopsis* [48]. Furthermore, among the major players in the photorespiration pathway, a central role is attributed to *HAO*, *CAT*, *rbcL*, *GLDC*, *AGXT*, and glyceric acid, which were highly accumulated in the N-use-inefficient genotypes under low N exposure, suggesting that they consumed more energy in photorespiration compared to the N-use-efficient ones.

### 3.5. Starch and Sucrose Metabolism

Sucrose is a raw material for many metabolic pathways, providing energy and carbon skeletons to macromolecules. Otherwise, starch plays a dual role in carbon allocation, acting as both a source, releasing carbon reserves in leaves for growth and development; and a sink, either as a dedicated starch store (seeds) or a temporary reserve of carbon contributing to sink strength in reproductive organs (flowers and fruits) [49].

In this pathway, different behaviors between the two pairs of NUE-contrasting genotypes were observed. Under N starvation (T_0_), the genes upstream of the fructose, glucose, and trehalose synthesis (*INV*, *GN4*, and *TPS*, respectively) were downregulated in the N-use-efficient compared to the inefficient genotypes, while sucrose synthase (*SUS)* and *endoglucanase* resulted in higher transcripts. Thus, the N-use-efficient genotypes utilized *SUS* to synthesize sucrose, while the inefficient ones hydrolyzed sucrose by invertase (encoded by *INV*) to produce glucose and fructose.

At T_1_, N-use-efficient genotypes exhibited a *WAXY* and *endoglucanase* (*E3.2.1.4*) upregulation, both playing a central role in the starch and amylose biosynthesis [50]. Endoglucanase contributes to the cellulose catabolic process during tissue development and cellulose degradation, making available the monosaccharides for consumption in chemical reactions [51]. By contrast, N-use-inefficient genotypes showed a higher sucrose accumulation, as well as *scrK*, *SUS*, and *TPS* transcript abundances. In detail, a high *TPS*-transcript abundance suggested a plant response to abiotic stress, causing a reduction in plant growth [52]. By contrast, the long-term low N exposure (T_2_) determined a *WAXY* and *endoglucanase* downregulation in the N-use efficient genotypes that in turns showed a *SUS* and *fructokinase* up-regulation as well as a higher fructose and glucose accumulation. It appears that the N-use-inefficient genotypes turn to cellulose degradation and starch biosynthesis (higher *WAXY*, *glgC*, and *endoglucanase* transcript accumulation was observed).

## 4. Materials and Methods

### 4.1. Plant Materials, Experimental Design, Tissue Sampling, and Sample Preparation

Two pairs of NUE-contrasting eggplants, named AM22, AM222, 67-3, and 305E40 were selected through hydroponic and greenhouse experiments [14]. In detail, AM222 and 67-3 were the N-use-efficient genotypes, while 305E40 and AM22 were the inefficient ones. Seeds, surface-sterilized with NaClO 5% (*v*/*v*) for 15 min and rinsed with deionized water, were germinated in Petri dishes (Ø 90 mm) on filter paper enriched with 0.1 mM CaSO_4_. After 10 days, seedlings with fully expanded cotyledons were selected and transferred to hydroponic tanks (4 L, ten seedlings per tank) containing 2.5 mM K_2_SO_4_, 2 mM MgSO_4_, 1 mM KH_2_PO_4_, 46 μM H_3_BO_3_, 9 μM MnCl_2_, 0.76 μM ZnSO_4_, 0.32 μM CuSO_4_, 0.11 μM Na_2_MoO_4_, 20 μM Fe-EDTA, and 4.75 mM CaSO_4_. The growing units were then transferred to a growth chamber at 24 °C, 65% relative humidity, and 14 h photoperiod with photon flux density of 350 µmol m^−2^ s^−1^ at plant height generated by high-pressure sodium discharge lamps. After two additional days, 0.5 mM NO_3_^−^ (as CaNO_3_) was added to the solution, and seedlings were grown for further 16 d. The nutrient solution was renewed every three days, and the pH was adjusted to 5.8 with 1 N KOH. Each genotype and tissue (root and shoot) were collected at T_0_ (before N supply), T_1_ and T_2_ (1 and 16 days after NO_3_^−^ resupply). Three biological replicates, consisting of eight bulked plants per replication for each tissue (root and shoot) after four hours of exposure to light were collected. The stored tissues were then powdered using an ice-cold mortar and pestle with liquid nitrogen for extraction of metabolites.

### 4.2. Metabolite Extraction and Annotation

The polar-metabolite extraction and derivatization for untargeted analysis by Gas Chromatography–Mass Spectrometry (GC-MS) using the method from Korenblum et al. [53] were carried out. The annotation was made by matching retention index and mass-spectrum data to the commercial Mass Spectral Library, NIST (www.nist.gov, 8 January 2022). The extraction of semipolar compounds for LCMS analysis by an ultra-high-performance liquid chromatography-quadrupole time-of-flight mass spectrometry (UPLC-qTOF-MS) (HDMS Synapt; Waters) was performed according to Itkin et al. [54] and the analyses were made according to Korenblum et al. [53]. Metabolites, comparing the retention times and the mass fragments to those of standard compounds injected at the same LCMS conditions, were identified. Compounds were putatively identified by comparing their retention times, elemental composition, and fragmentation pattern (MSE or ms/ms) with the home-made library, or as described in the literature when corresponding standards were not available.

### 4.3. RNAseq Analysis Data Validation by qRT-PCR

RNAseq analysis was previously carried out, and the most interesting identified DEGs validated by qRT-PCR [15]. Briefly, total RNA was isolated and purified using the Mini RNeasy Plant kit (QIAGEN, Milano, Italy), and 500 ng of total RNA per sample was used to construct cDNA libraries following the Transeq library procedures reported in Tzfadia et al. [55]. Libraries were sequenced on six lanes of HiSeq 2500 System (Illumina, San Diego, CA, USA), using the SR60 protocol. The Transeq output was ~3 million reads per sample. Resulting reads shorter than 30 bp were discarded and mapped to eggplant reference genome SMEL_V3.2016_11_01 from the Italian Consortium [56] using STAR vers. 2.4.2a (with EndToEnd option and outFilterMismatchNoverLmax was set to 0.04) [57]. RNAseq data were validated by using qRT-PCR analysis performed on a StepOnePlus Real-Time PCR System (Applied Biosystems, Life Technologies Corporation, Foster City, CA, USA), following the procedures reported by Mauceri et al. [14,15].

### 4.4. Statistical Analysis for Metabolite Profiling

Metabolomic analyses were performed in triplicate using a completely randomized experimental design, as reported in Section 4.1, and analyzed through the open-source software MetaboAnalyst 4.0 (www.metaboanalyst.ca, 9 January 2022) web [58]. Missing values were replaced by a small positive value (half of the minimum positive number detected in the data) and features with >50% missing values were removed. Then, raw data were normalized by a reference metabolite (ribitol), log10-transformed, and Pareto scaled [59]. The datasets were reduced by performing principal component analysis (PCA) and the partial least-squares discriminant analysis (PLS-DA). The model was validated and classified based on Q^2^ = 0.719 and R^2^ = 0.906. Twenty permutations with the *p*-value test < 0.05 were carried out. The score plots visualized the contrast between samples and the loading plots to explain the cluster separation with the variable importance of projection (VIP) score as cutoff ≥1. Data analyses were performed through the ANOVA univariate analysis using the least-significant difference (Fisher’s LSD) (*p* ≤ 0.05) as post hoc tests adjusted *p*-value (FDR) cutoff (≤0.05). To create a graphical heatmap with complete pairwise, a hierarchical clustering algorithm was adopted, and a Pearson correlation as distance measure was calculated.

### 4.5. KEGG Orthology (KO) Annotation and Transcriptomics and Metabolomics Integrated Correlation Network Analysis

RNAseq transcripts were functionally annotated using the Kyoto Encyclopaedia of Genes and Genomes (KEGG) database, and the R statistical package ‘Hmisc’v4.4-2 with Functions “rcorr” to estimate the Pearson Correlation Coefficient was used. Genes and metabolite-network analysis was carried out using MetScape v3.1 and Cytoscape v3.8.1, respectively. Correlation thresholds were defined using Pearson correlation coefficient (PCC) ≥0.70 and ≤−0.70 with *p*-value < 0.05. To obtain ENTREZ ID from symbol *Arabidopsis* ortholog of eggplant gene, a R-statistical package ‘org.At.tair.db’ v3.13 was employed. The integrated metabolic-pathway analysis among metabolomics and gene expressions was conducted by the Joint Pathway Analysis module (MetaboAnalyst 4.0). A graphical heatmap by the function “pheatmap” (package pheatmap) with the pairwise complete and Pearson method was created [60]. Pair comparisons were performed using volcano plot with FDR p-adjusted < 0.05) with LOG2 (FC) ≥ ±1 through statistical analysis module of the open-source software MetaboAnalyst 4.0 (www.metaboanalyst.ca, 9 Juanary 2022).

## 5. Conclusions

Multivariate analyses of primary and secondary metabolites contributed to a better understanding of NUE-contrasting eggplant plant responses. These approaches indicated that primary and secondary metabolites were affected by N stress in shoot and root, respectively. The analysis of these metabolites and their roles in each pathway showed that short- and long-term low N availability impacted the number and accumulation of specific classes of primary metabolites such as amino acids, sugars, and organic acids in the N-use-efficient genotypes.

Our study displayed differences among genotypes mainly in the shoot than in root; in detail, six different pathways appeared the most affected. Moreover, an integrated analysis between differential accumulated metabolites and expressed transcripts highlighted a central role of the glycine, serine, and threonine; glyoxylate and dicarboxylate; as well as starch and sucrose metabolisms. In the first two pathways, glycine and the related enzyme *glyA* seem to play a significant role in plant N-stress responses in the N-use-efficient genotypes. After two days of N starvation, an alternative higher accumulation of serine and glyceric acid in the N-use-inefficient genotypes was observed. A correlation between *SUS* and *fructokinase* transcript abundances and the D-glucose and D-fructose accumulation appeared useful to distinguish N-use-efficient and inefficient genotypes in starch and sucrose metabolism. Interestingly, at long-term low N exposure, a *WAXY* and *endoglucanase* downregulation in the N-use-efficient genotypes was evident, together with a *SUS* and *fructokinase* upregulation. By contrast, the N-use-inefficient genotypes turn towards cellulose degradation and starch synthesis (higher *WAXY*, *glgC*, and *endoglucanase* transcript accumulation was observed).

The responses observed in the N-use-efficient compared to the most inefficient genotype AM22 could represent a starting point for a deeper understanding of the mechanisms of eggplant adaptation to low N. Therefore, key transcripts and metabolites and their pathways unveiled in this study could be used as potential candidate targets for eggplant-NUE improvement.

## Figures and Tables

**Figure 1 plants-11-00700-f001:**
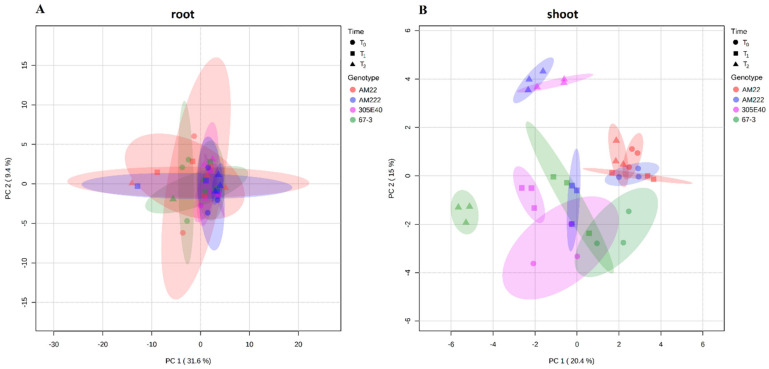
Two-dimensional plot of principal component analysis (PCA) of eggplant metabolites for root (**A**) and shoot (**B**). The dots represent accessions with 95% confidence regions as ellipses. (**A**) In root, PC1 and PC2 explained 41% of total variation; time sampling and accessions are not clearly distinguished. (**B**) In shoot, PC1 and PC2 explained 35.4% of total variations; AM22 do not respond to N limitation, while the other genotypes are clearly distinguished by treatments (time).

**Figure 2 plants-11-00700-f002:**
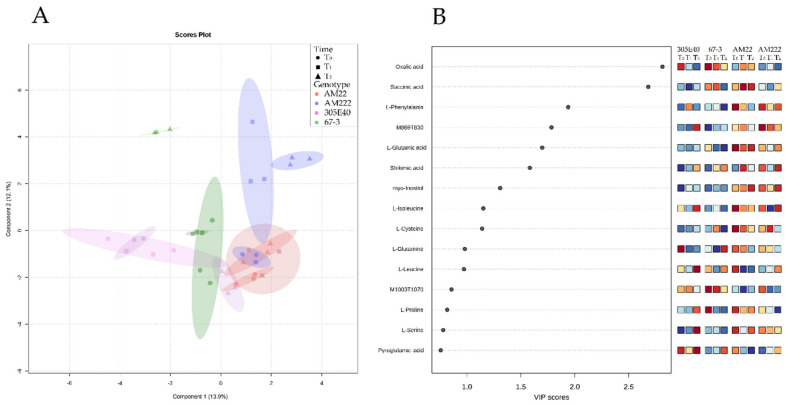
Partial least-squares discriminant analysis (PLS-DA) in shoot. (**A**) 2D scores plot of PLS-DA; the dots represent accessions with 95% confidence regions as ellipses. (**B**) The importance measures used in PLS-DA are VIP scores (variable importance in projection). The colored boxes on the right indicate the relative concentrations of the corresponding metabolite in each group.

**Figure 3 plants-11-00700-f003:**
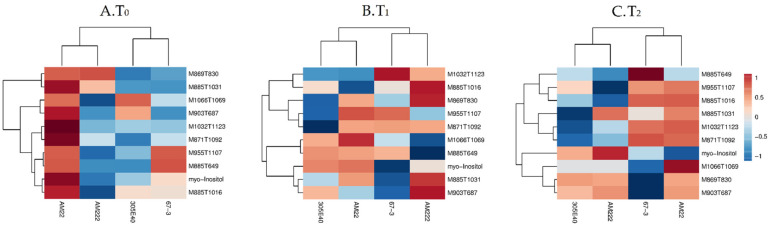
Heatmap of metabolites significantly differentially abundant among genotypes in root (one-way ANOVA and post hoc with *p* ≤ 0.05). Each column and row represent a sample and a metabolite, respectively. Comparison among genotypes shows that the main differences are in the secondary metabolites at T_0_ (**A**), T_1_ (**B**), and T_2_ (**C**).

**Figure 4 plants-11-00700-f004:**
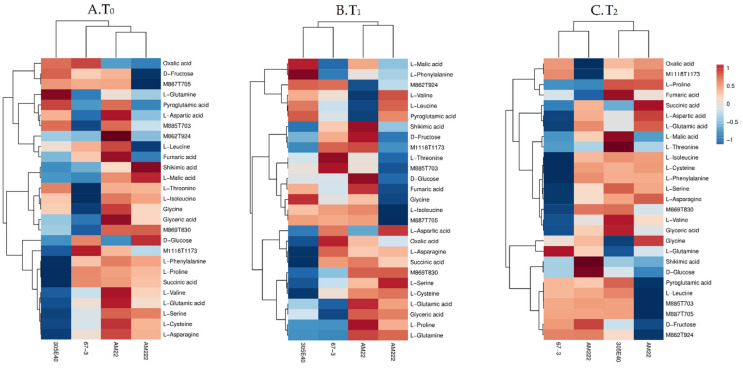
Heatmap of metabolites significantly differentially abundant among genotypes in shoot (one-way ANOVA and post hoc with *p* ≤ 0.05). Each column and row represent a sample and a metabolite, respectively. Comparison among genotypes shows that the main differences are in the primary metabolites at T_0_ (**A**), T_1_ (**B**), and T_2_ (**C**).

**Table 1 plants-11-00700-t001:** Comparative changes in the primary metabolite pathways in shoot. Metabolic pathways with FDR < 0.05 and higher impact values are highlighted. Pairwise comparison between genotypes. Total Cmpd represents the total compound number in the pathway; Hits is the actual matched number from the uploaded data; Raw p is the original *p* value calculated from the enrichment analysis; Holm p is the *p* value adjusted by Holm–Bonferroni method; FDR p is the *p* value adjusted using false discovery rate; Impact is the pathway impact value calculated from pathway topology analysis.

Pairwise Comparison in Shoot	Pathway Analysis	Total Cmpd	Hits	Raw p	−log(p)	Holm Adjust	FDR	Impact
T_0__67-3_*vs*_AM22	Alanine, aspartate, and glutamate metabolism	22	7	0.020155	1.6956	0.50388	0.047029	0.64748
T_0__305E40_*vs*_AM22				0.002014	2.6959	0.068486	0.0094	
T_1__AM222_*vs*_AM22				0.00278	2.5559	0.088969	0.010616	
T_1__67-3_*vs*_AM22				1.99 × 10^−5^	4.7012	0.00077607	0.000209	
T_1__305E40_*vs*_AM22				2.3 × 10^−5^	4.6388	0.00094188	0.00029	
T_2__AM222_*vs*_AM22	Starch and sucrose metabolism	22	2	0.003484	2.458	0.13935	0.038299	0.39104
T_2__67-3_*vs*_AM22	Alanine, aspartate, and glutamate metabolism	22	7	0.000618	3.2088	0.021642	0.003246	0.64748
T_2__305E40_*vs*_AM22	Glycine, serine, and threonine metabolism	33	5	0.000335	3.4751	0.013731	0.00559	0.53598
T_0__305E40_*vs*_AM222	Aminoacyl-tRNA biosynthesis	46	14	0.00032	3.4948	0.013443	0.013443	0.11111
T_1__305E40_*vs*_AM222	Alanine, aspartate, and glutamate metabolism	22	7	4.37 × 10^−5^	4.3594	0.0016609	0.000367	0.64748
T_2__305E40_*vs*_AM222				0.007058	2.1513	0.26116	0.049408	
T_1__67-3_*vs*_AM222	Alanine, aspartate, and glutamate metabolism	22	7	0.001201	2.9203	0.043252	0.007209	0.64748
T_2__67-3_*vs*_AM222	Phenylalanine metabolism	11	1	9.78 × 10^−5^	4.0098	0.0040085	0.000851	0.47059
T_0__67-3_*vs*_305E40	Glyoxylate and dicarboxylate metabolism	29	9	0.002673	2.5731	0.10691	0.037417	0.28209
T_1__67-3_*vs*_305E40	Alanine, aspartate, and glutamate metabolism	22	7	2.06 × 10^−5^	4.6871	0.00078109	0.000173	0.64748
T_2__67-3_*vs*_305E40				0.001274	2.8949	0.033116	0.003147	

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
