# Peer review of "Uncovering Pathways Highly Correlated to NUE through a Combined Metabolomics and Transcriptomics Approach in Eggplant"

_plants, 2022, doi:10.3390/plants11050700_

Round 1

Reviewer 1 Report

Nitrogen is one of the macronutrients of plants. The current paper determines the differential profiles of metabolites in root and shoot tissues, under N starvation as well as at short and long-term N limiting resupply using two eggplant genotypes with contrasting NUE. The results are interesting. However, some issues need to be addressed.

  1. Line 67: N starvation. Nitrogen has been abbreviated as N in the first appearance.
  2. Line 93: “thirty-three and twenty-nine” or “33 and 29”?
  3. Line 159-167: are there any other differences between the two genotypes? More detailed information is needed.
  4. Line 201: one hundred twenty-one or 121?
  5. What is the difference of metabolite changes between short and long- term N limiting condition as well as between two genotypes?
  6. What is the phenotypic and physiological responses of the two genotypes to N starvation?
  7. What is the reference gene for RNAseq analysis data validation by qRT-PCR?

Author Response

Nitrogen is one of the macronutrients of plants. The current paper determines the differential profiles of metabolites in root and shoot tissues, under N starvation as well as at short and long-term N limiting resupply using two eggplant genotypes with contrasting NUE. The results are interesting. However, some issues need to be addressed.

Line 67: N starvation. Nitrogen has been abbreviated as N in the first appearance.

Authors: thank you, it has been done.

Line 93: “thirty-three and twenty-nine” or “33 and 29”?

Authors: thank you for the suggestion, finally, we choice 33 and 29

Line 159-167: are there any other differences between the two genotypes? More detailed information is needed.

Authors: we do not understand what the reviewer means since in the sentences at lines 159-167 we included all the significant differences obtained; but many others are reported in the following sections. 

Line 201: one hundred twenty-one or 121?

Authors: thank you, it has been modified

What is the difference of metabolite changes between short and long- term N limiting condition as well as between two genotypes?

Authors: Thank you for your suggestion. We retain that the comparison between short and long-term could be not appropriate (T2 vs. T1). Since the plants were relatively young (T1), a difference of 16 days (T2), could be significantly affected by the growth stage, which in turn influenced greatly the metabolism and then the metabolite profile. For this reason, it would have been better to maintain in parallel control plants with an optimum nutrition and harvest at each time sampling. So, the plants can be compared at the same time and the differences would only been due to the N nutrition. We also analyzed the differences between genotypes (data not shown), but the results were not an added value to the paper, on the contrary they were redundant.

What is the phenotypic and physiological responses of the two genotypes to N starvation?

Authors: Thank you very much for your observation, which allowed us to detect a mistake/typo in the text. The starvation was only 2 days and needs to make negligible the nitrogen content in the seedling. All the seedlings were submitted to N starvation and after exposed to nitrate. Therefore, due to the starvation period, there is no phenotypic or physiological variation in the seedlings. T0 represents the starting point of experiment. I hope I was clear.

What is the reference gene for RNAseq analysis data validation by qRT-PCR?

Authors: thank you for your observation; the data validation was performed by two reference genes, the Adenine Phosphoribosyltransferase (APRT) and the Glyceraldehyde-3-Phosphate Dehydrogenase (GAPDH).

Reviewer 2 Report

The authors have integrated transcriptomics and metabolomics in eggplant to study its response to nitrogen starvation. This vegetable crop is less studied and such experiments will add new information to the database. The manuscript is well written and the experiment is well planned. I am not an expert in metabolomics methodology. The authors need to read the manuscript thoroughly and make minor corrections, some of which I have highlighted below.

Line 50: ‘Many efforts have been done’ change to ‘Efforts have been made’

Line 62: ‘contrasting genotype pairs’ change to ‘contrasting genotypes’

Line 78-79: reframe the sentence

Line 70-80: shift to ‘discussion’. Here, mention the objectives of the study.

Line 580: delete ‘by uniform size’; ‘ten seedlings for tank’ change to ‘ten seedlings per tank’

How nitrate was supplied, which salt was used?

Line 589: ‘eight bulked plants for replication’ change to ‘eight bulked plants per replication’

Line 610: cDNA libraries, following the Transeq; delete comma

Line 644: reframe the sentence

Line 87: 2.1. Metabolite detection in contrasting NUE genotypes and at different restoration time intervals

Line 258: ‘involved into the starch and sucrose’ change to ‘involved in the starch and sucrose’

Line 395: ‘regardless the genotype’ change to ‘regardless of the genotype’

Authors can also see the references which are very much relevant to their study and related to plant nutrition:

‘Nazir et al (2016) Nitrogen-Deficiency Stress Induces Protein Expression Differentially in Low-N Tolerant and Low-N Sensitive Maize Genotypes. Front. Plant Sci. 7:298. doi: 10.3389/fpls.2016.00298’

‘Ganie et al (2017) Nitrogen-regulated changes in total amino acid profile of maize genotypes having contrasting response to nitrogen deficit. Protoplasma (2017) 254:2143–2153; DOI 10.1007/s00709-017-1106-z’

‘Ganie et al. (2015) Metabolite Profiling of Low-P Tolerant and Low-P Sensitive Maize Genotypes under Phosphorus Starvation and Restoration Conditions. PLoS ONE 10(6): e0129520. doi:10.1371/journal.pone.0129520’

‘Ganie et al. Metabolite Profiling and Network Analysis Reveal Coordinated Changes in Low-N Tolerant and Low-N Sensitive Maize Genotypes under Nitrogen Deficiency and Restoration Conditions. Plants 2020, 9, 1459; doi:10.3390/plants9111459’.

Author Response

The authors have integrated transcriptomics and metabolomics in eggplant to study its response to nitrogen starvation. This vegetable crop is less studied and such experiments will add new information to the database. The manuscript is well written and the experiment is well planned. I am not an expert in metabolomics methodology. The authors need to read the manuscript thoroughly and make minor corrections, some of which I have highlighted below.

Line 50: ‘Many efforts have been done’ change to ‘Efforts have been made’

Authors: thank you for your suggestion, the sentence has been modified accordingly.

Line 62: ‘contrasting genotype pairs’ change to ‘contrasting genotypes’

Authors: thank you for your suggestion, the sentence has been modified accordingly.

Line 78-79: reframe the sentence

Authors: thank you for your suggestion, the sentence has been modified in “Under N stress, plants changed their metabolic profiles according to the starvation period [31], genotypes [32] and tissues.

Line 70-80: shift to ‘discussion’. Here, mention the objectives of the study.

Authors: thank you for the useful comment. We modified the sentence according to your suggestion retaining that as it is now has no longer the value of an obtained result, but adds informative value to the introduction.

Line 580: delete ‘by uniform size’; ‘ten seedlings for tank’ change to ‘ten seedlings per tank’

Authors: thanks for your comment. It has been modified as requested.

How nitrate was supplied, which salt was used?

Authors: thank you, we used calcium nitrate and we added the information in the material and method section at row 586.

Line 589: ‘eight bulked plants for replication’ change to ‘eight bulked plants per replication’

Authors: thanks for your comment. They have been modified as requested.

Line 610: cDNA libraries, following the Transeq; delete comma

Authors: the comma has been deleted.

Line 644: reframe the sentence

Authors: thanks so much for the suggestion, the sentence has been modified in “A graphical heatmap was create by the function “pheatmap” (package pheatmap) with the pairwise complete and method Pearson [88].”

Line 87: 2.1. Metabolite detection in contrasting NUE genotypes and at different restoration time intervals

Authors: thank you for suggesting this modification in the 2.1 subchapter. However, what do you think if we include “resupply” “2.1. Metabolite detection in contrasting NUE genotypes at different resupply time intervals”

Line 258: ‘involved into the starch and sucrose’ change to ‘involved in the starch and sucrose’

Authors: the “into” has been changed.

Line 395: ‘regardless the genotype’ change to ‘regardless of the genotype’

Authors: the sentence has been changed.

Authors can also see the references which are very much relevant to their study and related to plant nutrition:

‘Nazir et al (2016) Nitrogen-Deficiency Stress Induces Protein Expression Differentially in Low-N Tolerant and Low-N Sensitive Maize Genotypes. Front. Plant Sci. 7:298. doi: 10.3389/fpls.2016.00298’

‘Ganie et al (2017) Nitrogen-regulated changes in total amino acid profile of maize genotypes having contrasting response to nitrogen deficit. Protoplasma (2017) 254:2143–2153; DOI 10.1007/s00709-017-1106-z’

‘Ganie et al. (2015) Metabolite Profiling of Low-P Tolerant and Low-P Sensitive Maize Genotypes under Phosphorus Starvation and Restoration Conditions. PLoS ONE 10(6): e0129520. doi:10.1371/journal.pone.0129520’

‘Ganie et al. Metabolite Profiling and Network Analysis Reveal Coordinated Changes in Low-N Tolerant and Low-N Sensitive Maize Genotypes under Nitrogen Deficiency and Restoration Conditions. Plants 2020, 9, 1459; doi:10.3390/plants9111459’.

Authors: thank you very much for your suggestions. The papers suggested focused on maize, a species very far from the eggplant, therefore, among them we have retained useful to cite the last manuscript by Ganie et al 2020.

Reviewer 3 Report

In this manuscript, the authors did a very nice analysis with their data. I like read the paper. Please see my comments below,

Main comments

1, I would expect to see more original data in the main text. I do not like Table 1 and 2 presenting in the main text.

I will recommend the authors to bring figure S1 and S2 to the main text. Instead, bring Table 1 and 2 back to supplemental data.

2, Figure legends should be more detail to explain all the information present in the figure.

Other comments

1, line 58, remove “and in model plants”.

2, line 587, I think it should be “after two days N free solutions”.

Author Response

In this manuscript, the authors did a very nice analysis with their data. I like read the paper. Please see my comments below.

Main comments

1, I would expect to see more original data in the main text. I do not like Table 1 and 2 presenting in the main text. I will recommend the authors to bring figure S1 and S2 to the main text. Instead, bring Table 1 and 2 back to supplemental data.

Authors: thanks for your suggestions. In agreement, table 1 and 2 have been moved as supplementary tables, while figure S1 e S2 have been included in the manuscript as figures 3 and 4 as suggested by the reviewer.

2, Figure legends should be more detail to explain all the information present in the figure.

Authors: The legends of the figures have been improved as suggested and the captions have been more detailed.

Other comments

1, line 58, remove “and in model plants”.

Authors: it has been deleted.

2, line 587, I think it should be “after two days N free solutions”.

Authors: it has been changed.

Reviewer 4 Report

Comments on plants-1602087

The authors compared two pairs of NUE-contrasting eggplant (Solanum melongena L.) genotypes, employing GC-MS and UPLC-qTOF-MS-based technologies to determine the differential profiles of primary and secondary metabolites in root and shoot tissues, under N starvation as well as at short and long-term N limiting resupply.

Overall, the paper is written well

The treatments abbreviation should be properly written using subscripts with the numbers throughout the manuscript

Regarding the references consulted and cited, these should be reduced to max. 60 by keeping the most relevant and recent. No paper has been consulted from 2022; there should be some references from the recent literature  

Author Response

The authors compared two pairs of NUE-contrasting eggplant (Solanum melongena L.) genotypes, employing GC-MS and UPLC-qTOF-MS-based technologies to determine the differential profiles of primary and secondary metabolites in root and shoot tissues, under N starvation as well as at short and long-term N limiting resupply.

Overall, the paper is written well

The treatments abbreviation should be properly written using subscripts with the numbers throughout the manuscript

Authors: thanks for your suggestion. Although it could be plausible to utilize the subscript for the numbers included in the treatment abbreviation, we would prefer to maintain this abbreviation in agreement with the previous papers already published (Mauceri et al 2020; 2021).

Regarding the references consulted and cited, these should be reduced to max. 60 by keeping the most relevant and recent. No paper has been consulted from 2022; there should be some references from the recent literature 

Authors: thanks for your suggestion. Therefore, I would remember to the reviewer that the manuscript has been submitted in January 2022 and we have cited many papers by 2020 and 2021. Regarding the number of citations, we refer to the rule of the journal where there is not any limitation.

Round 2

Reviewer 4 Report

The authors have not still incorporated the suggested changes. Regarding the reasons for not incorporation of suggested changes, it is not good or recommended to follow a mistake/s as it/these were done by the authors themselves in their previous papers or others. The authors are advised to modify as per the following corrections suggested 

The treatments abbreviation are required to be written using subscripts with the numbers throughout the manuscript

For a research paper, max. 60 references, the most suitable ones are sufficient to justify a research work. Of course, the authors have submitted their manuscript in January but now, it is February 19, 2022. The authors are highly suggested to consult a few papers from 2022 so that there would be updated literature added in the Introduction and Discussion sections. 

Author Response

The authors have not still incorporated the suggested changes. Regarding the reasons for not incorporation of suggested changes, it is not good or recommended to follow a mistake/s as it/these were done by the authors themselves in their previous papers or others. The authors are advised to modify as per the following corrections suggested 

The treatments abbreviation are required to be written using subscripts with the numbers throughout the manuscript

Authors: Thank you, as you requested the treatment symbols have been changed using subscripts for the numbers throughout the manuscript and in the figures and relative legends.

For a research paper, max. 60 references, the most suitable ones are sufficient to justify a research work. Of course, the authors have submitted their manuscript in January but now, it is February 19, 2022. The authors are highly suggested to consult a few papers from 2022 so that there would be updated literature added in the Introduction and Discussion sections. 

Authors: Thank you, as suggested we have re-edited the references list, including in both introduction and discussion new very recent papers (2022), and deleting the oldest redundant references, some of these replaced by the recent ones (2022). We have reached the number of citations requested (60) that well sustain the manuscript.

Round 3

Reviewer 4 Report

The authors have incorporated all the suggested changes. However, the quality/visibility of the revised Figures 2-4 is very low and needs attention. The Figures 2-4 must be replaced with high-resolution ones in the final version so that these could easily be observed 

Author Response

Thank you so much, we have replaced the figures 2-4 as requested.
